# Bet-hedging antimicrobial strategies in macrophage phagosome acidification drive the dynamics of *Cryptococcus neoformans* intracellular escape mechanisms

**Quigly Dragotakes**[1], **Ella Jacobs**[1], **Lia Sanchez Ramirez**[2], **Olivia Insun Yoon**[2], **Caitlin Perez-Stable**[2], **Hope Eden**[3], **Jenlu Pagnotta**[2], **Raghav Vij**[4], **Aviv Bergman**[5,6], **Franco D'Alessio**[7], **Arturo Casadevall**[1]*

1 Department of Molecular Microbiology and Immunology, Johns Hopkins Bloomberg School of Public Health, Baltimore, Maryland, United States of America, 2 Department of Molecular and Cell Biology, Johns Hopkins University, Baltimore, Maryland, United States of America, 3 Department of Cellular and Molecular Medicine, Johns Hopkins University School of Medicine, Baltimore, Maryland, United States of America, 4 Leibniz Institute for Natural Product Research and Infection Biology, Hans Knöll Institute, Jena, Germany, 5 Department of Systems and Computational Biology, Albert Einstein College of Medicine, New York City, New York, United States of America, 6 Santa Fe Institute, Santa Fe, New Mexico, United States of America, 7 Department of Medicine, Johns Hopkins University School of Medicine, Baltimore, United States of America

* acasade1@jhu.edu

**Data Availability Statement:** All relevant data are within the manuscript and its Supporting Information files.

## Abstract

The fungus *Cryptococcus neoformans* is a major human pathogen with a remarkable intracellular survival strategy that includes exiting macrophages through non-lytic exocytosis (Vomocytosis) and transferring between macrophages (Dragotcytosis) by a mechanism that involves sequential events of non-lytic exocytosis and phagocytosis. Vomocytosis and Dragotcytosis are fungal driven processes, but their triggers are not understood. We hypothesized that the dynamics of Dragotcytosis could inherit the stochasticity of phagolysosome acidification and that Dragotcytosis was triggered by fungal cell stress. Consistent with this view, fungal cells involved in Dragotcytosis reside in phagolysosomes characterized by low pH and/or high oxidative stress. Using fluorescent microscopy, qPCR, live cell video microscopy, and fungal growth assays we found that the that mitigating pH or oxidative stress reduced Dragotcytosis frequency, whereas ROS susceptible mutants of *C. neoformans* underwent Dragotcytosis more frequently. Dragotcytosis initiation was linked to phagolysosomal pH, oxidative stresses, and macrophage polarization state. Dragotcytosis manifested stochastic dynamics thus paralleling the dynamics of phagosomal acidification, which correlated with the inhospitality of phagolysosomes in differently polarized macrophages. Hence, randomness in phagosomal acidification randomly created a population of inhospitable phagosomes where fungal cell stress triggered stochastic *C. neoformans* non-lytic exocytosis dynamics to escape a non-permissive intracellular macrophage environment.

**Funding:** This work was supported by the National Institutes of Health (R01HL059842, 5R01AI033774, 5R37AI033142, and 5R01AI052733 to AC). The funders had no role in study design, data collection and analysis, decision to publish, or preparation of the manuscript.

**Competing interests:** The authors have declared that no competing interests exist.

## Author summary

Host macrophages do not have prior information about the pathogens they ingest. Hence, they have no information about the threats from or vulnerabilities of microbes they phagocytose. Consequently, macrophages must hedge bets in their defense strategies when combatting pathogens with unknown vulnerabilities. This results in a wide variety of phagosomal environments which maximizes the overall antimicrobial ability of the entire population of phagosomes but individual phagosomes may or may not be hostile to a specific pathogen. Using live cell microscopy to track in vitro infections, viability staining, qPCR, and enzyme activity assays we investigated the relationship between phagosomal cell stress and Dragotcytosis, or fungal macrophage-to-macrophage transfer. We have found that *Cryptococcus neoformans* game this system by transferring to new phagosomes when it finds itself in too hostile of an environment. Specifically, this process is triggered when the yeast experiences overwhelming cellular stress. These findings illustrate a system in which *C. neoformans* is detecting cellular damage as a trigger for the transfer process and the stochastic nature of microbe-macrophage interactions at the cellular level.

## Introduction

*Cryptococcus neoformans* is a pathogenic yeast that can reside in the phagolysosome of host macrophages [1]. Macrophages are critical cells in the pathogenesis of cryptococcosis, being involved in the containment of, and extrapulmonary dissemination of infection [2]. Macrophages are also involved in the pathogenesis of latent infection where the organism can survive for a long time in granulomas. Hence, *C. neoformans* survival in macrophages is critical for both the persistence and dissemination of infection. The yeasts, inhaled from the environment, manage intracellular survival by modifying the phagolysosomal environment in their own favor. The capsule of *C. neoformans* is comprised of several subunits including glucuronic acid and glucuronoxylomannan (GXM) which act as weak acids capable of buffering the environment to pH ~5 [3,4]. *C. neoformans* also produces urease which disrupts phagolysosomal acidification by breaking down urea into carbon dioxide and ammonia, a weak base [5]. Additionally, *C. neoformans* are capable of exiting host macrophages through lytic egress, non-lytic exocytosis (Vomocytosis) [6–8], or lateral cell-to-cell transfer (Dragotcytosis) [9].

We recently observed that macrophages use a bet hedging strategy in phagosomal acidification to maximize the antimicrobial properties of acidic pH for controlling the growth of ingested microbes [10]. By increasing the diversity of possible phagolysosomal pH, a population of macrophages will optimize its chances to inhibit pathogen growth based on pH alone. We also noticed that even small perturbations in this system could disrupt the bet-hedging strategy and that different pathogens employ various strategies to tip the odds in their own favor. *C. neoformans* interferes with the stochastic modulation of phagosomal pH by capsule buffering and urease activity [3,5]. Phagosomal pH varies with macrophage polarization such that M2 polarized macrophages are less hostile [10], in terms of pH, to ingested *C. neoformans* yeasts.

A fascinating aspect of the interaction with *C. neoformans* with macrophages is the phenomenon of fungal cell transfer between two macrophages, a process we have recently termed 'Dragotcytosis' [9]. Dragotcytosis results from coordinated sequential exocytosis and phagocytosis events between adjacent macrophages [9]. We also noted that Dragotcytosis is favorable to *C. neoformans* as macrophages with Dragotcytosis blockaded yield fewer colony forming units after 24 h than those who were not blockaded. However, Dragotcytosis is also potentially

beneficial to the host cell if the continued fungal intracellular residence would result in more cellular damage. Non-lytic exocytosis events avoid lytic escape, resulting in fewer lysed host cells, a type of relationship observed in other amoeba ingested microbes like *Mycobaterium marinum* [11,12]. Finally, we note that Dragotcytosis is an active process, only occurring with live *C. neoformans* [6]. When taken together, these findings suggest Dragotcytosis is triggered by *C. neoformans* for a purpose beneficial to the yeast. However, it is unknown what specifically triggers Dragotcytosis or how transferring between macrophages confers a benefit Notably, previous observations have established that phagolysosomal pH can modulate non-lytic exocytosis frequency, supporting the idea that pH has an important and complex role in the regulation of this system [13,14].

In addition to phagolysosomal pH there are other stressors in the phagosome that could conceivably affect the rate of Dragotcytosis. The oxidative burst and the release of reactive oxygen species into the phagolysosome are also potentially important. Both ROS generation and nitric oxide synthase (NOS) activity are upregulated in M1 macrophages in response to phagocytosis [15–17]. NOS metabolizes arginine into nitric oxide and citrulline. Conversely, Arginase (Arg), upregulated in M2 macrophages) hydrolyzes arginine to ornithine and urea. Increased activity of either Arg or NOS will consume available arginine, resulting in lower efficacy of the other. Thus, M2 polarized macrophages have lower NOS activity and a less significant oxidative burst, which is thought to be one of the main reasons M2 macrophages are less effective at killing pathogens.

In this study, we analyzed the consequences of macrophage polarization and phagolysosomal stress on *C. neoformans* pathogenesis and outcome. We hypothesized that macrophages containing *C. neoformans* phagolysosomes that caused significant stress to the yeast by acidity and/or oxidative stress would be more likely to trigger Dragotcytosis events than other *C. neoformans* containing phagolysosomes. In this regard, we tested whether the more fungicidal environment in M1 macrophages was more likely to trigger Dragotcytosis than the more permissive intracellular environment of M2 macrophages. Our goal was evaluating the hypothesis that Dragotcytosis is a fungal driven mechanism for escape of a hostile environment. We found that both hostile pH and reactive oxygen species (ROS) in the phagolysosome drive Dragotcytosis and use modelling to show the potential increasing benefit of Dragotcytosis events as a function of resident phagolysosome hostility.

## Materials and methods

### Ethics statement

Murine experiments were carried out in accordance with approved IACUC guidelines (protocol MO21H124). Mice were sacrificed using $CO_2$ asphyxiation.

### Cell strains and culture conditions

*Cryptococcus neoformans* species complex serotype A strain H99 was originally obtained from John Perfect (Durham, NC) and *C. neoformans* H99 actin-GFP created by the May lab [18]. Culture stocks were stored at 80˚C. Frozen stocks were later streaked onto YPD agar and incubated at 30˚C. Liquid suspensions of cryptococcal cultures were grown in YPD overnight at 30˚C. Cryptococcal cultures were heat killed by incubating at 65˚C for 1 h. Mutant strains were obtained from a previously published knockout library [19].

Bone marrow derived macrophages (BMDMs) were harvested from 6-week-old C57BL/6 or B6.129S-*Cybb*[tm1Din]/J female mice from The Jackson Laboratory from hind leg bones and were differentiated by seeding in 10 cm tissue culture treated dishes in DMEM with 10% FBS, 1% nonessential amino acids, 1% penicillin-streptomycin, 2 mM Glutamax, 1% HEPES buffer,

20% L-929 cell conditioned supernatant, 0.1% beta-mercaptoethanol for 6 days at 37°C and 9.5% $CO_2$. BMDMs were used for experiments within 5 days after differentiation. BMDMs were activated with 0.5 ug/mL LPS and 10 ng/mL IFN-γ for M1 polarization or 20 ng/mL IL-4 for M2 polarization for 16 h prior to experiments.

## Phagolysosomal pH measurement

Phagolysosomal pH was measured using ratiometric fluorescence imaging involving the use of pH-sensitive probe Oregon green 488 as described in prior studies [5]. Briefly, Oregon green 488 was first conjugated to mAb 18B7 using Oregon Green 488 Protein Labeling Kit (Thermo-Fisher). BMDMs were plated at a density of $1.25 \times 10^5$ cells/well on 24-well plate with 12 mm circular coverslip. Cells were activated with 0.5 μg/ml LPS and 100 U/ml IFN-γ or 20 ng/mL IL-4 as previously described at 37°C in a 9.5% $CO_2$ atmosphere overnight. Prior to infection, 2 d old live H99, heat killed H99, or anti-mouse IgG coated polystyrene bead ($3.75 \times 10^6$ cells or beads/ml) were incubated with 10 μg/ml Oregon green conjugated mAb 18B7 for 15 min. Macrophages were then incubated with Oregon green conjugated mAb 18B7-opsonized particles in $3.75 \times 10^5$ cryptococcal cells or beads per well. Cells were either centrifuged immediately at 350 x $g$ for 1 min and incubated at 37°C for 10 min or incubated at 4°C for 30 min to synchronize ingestion and allow phagocytosis. Extracellular cryptococcal cells or beads were removed by washing three times with fresh medium, a step that prevents the occurrence of new phagocytic events. As an additional safeguard against new phagocytic events fresh media was supplemented with AlexaFluor 568 conjugated mAb 18B7 for 1 h to label extracellular particles. Samples were collected at their respective time points after phagocytosis by washing the coverslip twice with pre-warmed HBSS and placing it upside down on a MatTek petri dish (35 mm with 10 mm embedded coverslip well) with HBSS in the microwell. Images were taken by using Olympus AX70 microscopy with objective 40x at dual excitation 440 nm and 488 nm, and emission 520 nm. Images were analyzed using MetaFluor Fluorescence Ratio Imaging Software. Fluorescence intensities were used to determine the ratios of Ex488 nm/Ex440 nm that were converted to absolute pH values using a standard curve where the images are taken as above but intracellular pH of macrophages was equilibrated by adding 10 μM nigericin in pH buffer (140 mM KCl, 1 mM $MgCl_2$, 1 mM $CaCl_2$, 5 mM glucose, and appropriate buffer ≤ pH 5.0: acetate-acetic acid; pH 5.5–6.5: MES; ≥ pH 7.0: HEPES. Desired pH values were adjusted using either 1 M KOH or 1 M HCl). The pH of buffers was adjusted at 3–7 using 0.5-pH unit increments.

## Cryptococcal capsule measurements

Capsule measurements were acquired by measuring exclusion zones on India Ink slides and phase contrast microscopy. To determine differences between polarized macrophage incubations, *C. neoformans* were harvested after being ingested by macrophages for 24 h. Extracellular yeasts were first removed by washing the cells 3 times with 1 mL HBSS. Macrophages were lifted from their plates, centrifuged at 350 $g$ for 10 min, and resuspended in 1 mL distilled $H_2O$. Cells were then passed through a 27 ¾ gauge needle 10 times and left incubating for 20 total min to ensure lysis. After lysis *C. neoformans* were pelleted via centrifugation at 2300 $g$ for 5 min and resuspended in 50 μL of PBS. Slides were prepared using 8 μL of cell mixture and 1.5 μL India ink, then imaged on an Olympus AX70 at 20x objective. Capsules and cell bodies were measured using a previously published measuring program [20].

## NOS and arginase activity measurements

BMDMs were seeded at $10^6$ cell/well in 6-well treated tissue culture plates and activated overnight for M0, M1, or M2 polarization as previously described. Prior to infection, 2 d old live

H99 ($10^6$ cells/well) were incubated with 10 μg/ml mAb 18B7 for 10 min. Macrophages were then incubated with opsonized particles at MOI of 1. Cells were either centrifuged immediately at 350 *g* for 1 min or incubated at 4°C for 30 min to synchronize ingestion and cultures were incubated at 37°C for 10 min to allow phagocytosis. Extracellular cryptococcal cells were removed by washing three times with fresh medium, a step that prevents the occurrence of new phagocytic events. After 24 hours, cell supernatant was collected and the BMDMs were lysed with distilled water and 10 passages through a syringe with 23G needle. The supernatant was tested for NO levels using Greiss reagent kit (Millipore-Sigma G4410). Cell lysates were tested for arginase activity with arginase activity assay kit (Millipore-Sigma MAK112).

### *Cryptococcus neoformans* viability and growth inhibition

BMDMs were seeded ($10^6$ cells/well) in 6-well tissue culture plates. The cells were activated overnight (16 h) with IFNγ (0.02 μg/mL) and (0.5 μg/mL) LPS for M1, IL-4 (20 ng/mL) for M2, or unstimulated for M0. The cells were then infected with a 2 d culture of *C. neoformans* opsonized with 18B7 (10 μg/mL) at an MOI of 1 for 2 or 24 h. At the respective timepoint, the supernatant was collected and total yeast cells as well as GFP positive yeast were counted by hemocytometer and fluorescence microscopy using an Olympus AX70. *C. neoformans* ingested by adherent macrophages were imaged on a Zeiss Axiovert 200M inverted scope in a live cell incubator chamber. Viability was calculated as the proportion of yeast cells expressing GFP-actin. A second measure of viability was acquired by staining collected yeast on a hemocytometer in 1:1 0.4% trypan blue solution (Sigma Aldrich) and enumerating total blue cells compared to total cells.

The collected, conditioned supernatant was then pelleted at 2300 x *g* for 5 min to remove debris and extracellular crypto. Supernantants were seeded with 5 x $10^4$ cell / mL of *C. neoformans* and absorbance (600 nm) as well as fluorescence (GFP 488/510) was read at 0 and 24 h. Samples were prepared on glass slides with India Ink negative stain and imaged with an Olympus AX70. Growth was characterized by increase in OD600 and viability was measured as the proportion of GFP positive cells.

### *Cryptococcus neoformans* growth curve

*C. neoformans* strain H99 was seeded in 3 mL of YPD media for 2 d at 30 °C on a 120 rpm rotational wheel. Cultures were counted via hemocytometer and seeded into a 24 well plate at $10^4$ cell/mL in 1 mL total volume of YPD. Experimental wells were supplemented with bafilomycin A1 (100 nM), chloroquine (6 μm), fluconazole (20 μg / μL), amphotericin B (0.5 μg / μL), or 1400W (100 μM). Absorbance (600 nm) was then read every 5 min for 3 d by spectrophotometer (Spectramax M5) at 30 °C or 37 °C with shaking between measurements.

### Dragotcytosis frequency measurements

BMDMs were seeded (5 x $10^4$ cells/well) in MatTek dishes. The cells were activated overnight (16 h) with IFNγ (0.02 μg/mL) and (0.5 μg/mL) LPS for M1, IL-4 (20 ng/mL) for M2, unstimulated for M0, or stimulated M1 and supplemented with 1400W (100 μM). Cells in the MatTek dish were infected with Uvitex 2B (5 μm/mL) stained and 18B7 (10 μg/mL) opsonized *C. neoformans* at an MOI of 3 for 1 h, then supplemented with 2 mL fresh media and 18B7 mAb. In the case of drug trials this fresh media was also supplemented with bafilomycin A1 (100 nM), chloroquine (6 μm), fluconazole (20 μg / μL), or amphotericin B (0.5 μg / μL). MatTek dishes were then placed under a Zeiss axiovert 200M 10X magnification, incubated at 37°C or 30°C and 9.5% $CO_2$, and imaged every 2 min for a 24 h period. Images were then manually analyzed to identify ingested yeast cell outcomes.

## Modelling

To simulate the effect of Dragotcytosis on a population of *C. neoformans* in phagolysosomes we generated 10,000 hypothetical phagolysosomal pH values based on the distribution of observed phagolysosomal pH in M1 polarized bead containing macrophage phagolysosomes. Each value < 4 was replaced one time by randomly determining a new phagolysosomal pH from the same distribution to simulate a Dragotcytosis event from the initial macrophage to a random new one.

## Data analysis for stochastic signatures

Discrimination of deterministic vs. stochastic dynamics was achieved using the previously characterized permutation spectrum test [21] and the methods outlined in our previous work [10]. In short, processed datasets are aligned in a vector and separated into subgroups by a scanning window of 4 units. Each of the overlapping four-unit segments are then assigned a value from 0 to 3 based on their relative size (the largest value being 3 and the least being 0) in what is referred to as an ordinal pattern. The frequencies for each pattern are calculated for the entire dataset. Determinism is characterized by observations of ordinal patterns that do not exist (forbidden patterns), whereas stochastic dynamics are characterized by every ordinal pattern existing at a non-zero frequency. In this work, the original data is the time interval between the initiation of each instance of host cell escape within an experiment.

## qPCR

BMDMs were seeded at $10^6$ cell/well in 6-well treated tissue culture plates and activated overnight for M0, M1, or M2 polarization as previously described. Prior to infection, 2 d old live H99 or inert beads ($10^6$ particles/well) were incubated with 10 µg/ml mAb 18B7 for 10 min. Macrophages were then incubated with opsonized particles at MOI 1. Cells were either centrifuged immediately at 350 x *g* for 1 min or incubated at 4˚C for 30 min to synchronize ingestion and cultures were incubated at 37˚C for 10 min to allow phagocytosis. Extracellular cryptococcal cells were removed by washing three times with fresh medium, a step that prevents the occurrence of new phagocytic events. After 24 hours, BMDMs were resuspended in TRIzol reagent and frozen at -80˚C. Total RNA was isolated from frozen cell cultures using the TRIzol reagent per the manufacturer's suggestions (InVitrogen, Carlsbad, CA). The RNA was precipitated in isopropanol and then subjected to additional purification using the RNeasy RNA isolation kit (Qiagen, Valencia, CA) according to the manufacturer's instructions. RNA was reverse transcribed to generate cDNA using a First-Strand Synthesis kit (Amersham) and random hexamers as primers. The cDNA was then used as the template for quantitative PCR using an iCycler Thermal Cycler Real-Time PCR machine (Bio-Rad, Hercules, CA). The products of PCR amplification were detected with Syber green fluorescent dye and the relative expression of each gene of interest expressed with reference to that of glyceraldehyde phosphate dehydrogenase (GAPDH). The PCR products were analyzed on Tris-Acetate-EDTA agarose gels to confirm its correct size. Analysis was performed with commercially available and optimized Qiagen primers (Table 1).

## Macrophage preference assay

To ascertain whether *C. neoformans* preferred residence in macrophages as a function of host cell polarization we carried out cell transfer experiments in macrophage populations consisting of mixed M0, M1, and M2 polarized cells. BMDMs were seeded at 3 x $10^4$ cells per well in a 24 well tissue culture plate loaded with circular coverslips then activated overnight for M1

**Table 1. qPCR primer metadata.**

| Species | Gene name | Catalog number/ GeneGlobe ID | Amplicon Length | Detected Transcripts |
|---|---|---|---|---|
| Mouse (Mus musculus) | Arg-1 (arginase, liver) aka Arginase-1 | PPM31770C-200 | 73 | NM_007482 (1489 bp) |
| Mouse (Mus musculus) | IL-12b (interleukin 12b) | PPM03020E-200 | 114 | NM_001303244 (2505 bp) |
| Mouse (Mus musculus) | Chil3 (chitinase-like 3) | PPM25130A-200 | 89 | NM_009892 (1567 bp) |
| Mouse (Mus musculus) | Nos2 (nitric oxide synthase 2, inducible) | PPM02928B-200 | 122 | NM_001313921 (3915 bp) |
| Mouse (Mus musculus) | TNF-a (Tumor Necrosis Factor) | PPM03113G-200 | 153 | NM_013693 (1653 bp) |
| Mouse (Mus musculus) | Ret (ret proto-oncogene) | PPM05472F-200 | 65 | NM_001080780 (7341 bp) |
| Mouse (Mus musculus) | GAPDH (glyceraldehyde-3-phosphate dehydrogenase) | PPM02946E-200 | 140 | NM_008084 (1444 bp) |

polarization as previously described. Three separate populations of BMDMs were seeded in dishes large enough to account for an additional $3 \times 10^4$ cells per well per condition and activated overnight for M0, M1, or M2 polarization as previously described. The initial population of M1 BMDMs on coverslips was infected with $6 \times 10^4$ 18B7 opsonized *C. neoformans* yeasts per well for 1 h. The remaining three populations of uninfected BMDMs were stained with CellTracker CMFDA (Green). After the 1 h infection the three labeled populations of BMDMs were lifted with cellstripper and added to the unlabeled, infected M1 coverslips so that each coverslip had two populations of macrophages: 1. Unlabeled and infected M1 and 2. Labeled but uninfected M0, M1, or M2. The coverslips were then incubated for 24 h at 37°C and 9.5% $CO_2$ for 24, 48, or 72 h. The cells were then fixed with 4% paraformaldehyde for 10 min at room temperature and coverslips were mounted on slides with ProLong Gold Antifade Mountant (Thermofisher) and imaged on an Olympus AX70. For each field of view the total number of each macrophage population was counted along with how many infected macrophages in each population. To correct for floating cells and ensure a proper count of total cells, a hemocytometer was used to enumerate the total number of floating cells in each sample. We assumed any floating cells were previously evenly distributed across the original sample and thus used the percent of floating cells to estimate the number of missing cells in any given sample using the formula below:

$$T'_G = \left[ \frac{T_G}{1 - \frac{SN_G}{I_G}} \right]$$

$T_G$ = Total green positive cells counted on a given field of view
$SN_G$ = Total number of green cells in supernatant per hemocytometer
$I_G$ = Total amount of green cells plated initially

### Modeling dragotcytosis dynamics

A population of 40,000 cells was generated consisting of 10,000 cells each of infected M1, uninfected M1, uninfected M2, and uninfected M0 macrophages. During each iteration of Dragotcytosis each infected cell had a random chance of experiencing Dragotcytosis or Vomocytosis based on observed frequencies *in vitro*. A Vomocytosed yeast was considered removed from the modeling experiment as we did not have an accurate way to model reuptake, and a Dragotcytosed yeast was randomly transferred to one of the four types of macrophage cells with no bias introduced. 10 rounds of sequential Dragotcytosis were modeled and the entire computational experiment was replicated 100 times.

### Statistical analysis

Specific statistical tests for each experiment are denoted in the figure legends with tests for multiple hypotheses. Graphical error bars indicate 95% confidence intervals of the respective measurement. When comparing frequency of host cell exit strategies, we compared mutant strains to the wild-type strain with a test of equal proportions and Bonferroni multiple hypothesis correction.

## Results

### Macrophage polarization state affects frequency of dragotcytosis uncoupled from vomocytosis

Given our prior findings showing different acidification dynamics in differently polarized macrophages [10], we investigated whether macrophage polarization effected the frequency of Dragotcytosis. First, to ensure that our results were not confounded by differing population densities of macrophages resulting in higher or lower frequencies of events, we compared Dragotcytosis frequency to macrophage density and found that while the total number of Dragotcytosis events increases with total number of macrophages, as expected, there was no correlation between the frequency of events overall (**S1 Fig**). Vomocytosis and Dragotcytosis frequencies in macrophages infected with *C. neoformans* strain H99 correlated (Pearson correlation 0.41 with $P = 0.055$ and Spearman correlation 0.48 with $P = 0.025$), consistent with the notion that both processes are coupled (**S1D Fig**). Analysis of the pH distribution in cryptococcal phagosomes, M2 macrophages are more hospitable that M1 macrophages *C. neoformans*, having a higher proportion of macrophages at optimal growth pH (~5.5) and few phagolysosomes at inhibitory ranges ($\leq 4$) (**Fig 1**A). Using M0 macrophages as a baseline measure of Dragotcytosis frequency, we find that M1 polarization increases Dragotcytosis frequency while M2 polarization completely abrogates Dragotcytosis. (**Figs 2 and S2**). This result correlates the frequency of Dragotcytosis with the relative hospitality of each type of macrophage phagolysosome. When taken in context with the previously reported insight that Dragotcytosis is an active process which is beneficial for *C. neoformans* survival [9], these data suggest that Dragotcytosis is more likely to be initiated in hostile phagolysosomes.

### *Cryptococcus neoformans* Capsule and cell body size are unaffected by polarization

The *C. neoformans* capsule is one of the most important virulence factors and determinants of infection outcome and a powerful modulator of phagolysosomal pH [3]. Capsule growth could modulate phagolysosomal pH by mechanically stressing the phagolysosomal membrane as it enlarges, causing the phagolysosome to become leaky [22]. Several aspects of cryptococcal pathology and phagolysosomal outcome are correlated to capsule size including buffering potential [3,22]. To investigate whether the changes in pH in differently polarized macrophages were due to capsule differences, we measured capsule sizes after infection of differently polarized macrophages. We found that the average and median capsule sizes did not appreciably differ between populations of *C. neoformans* ingested by differently polarized macrophages, nor did the average and median cell body sizes (**S3 Fig**). Furthermore, if phagolysosomal pH was modulated by capsule size, we would expect to observe a correlation between cell radius and phagolysosomal pH. We found no indication of a correlation between particle size and phagolysosomal pH, even if we began measuring clusters of multiple particles within a single phagolysosome (**S4 Fig**).

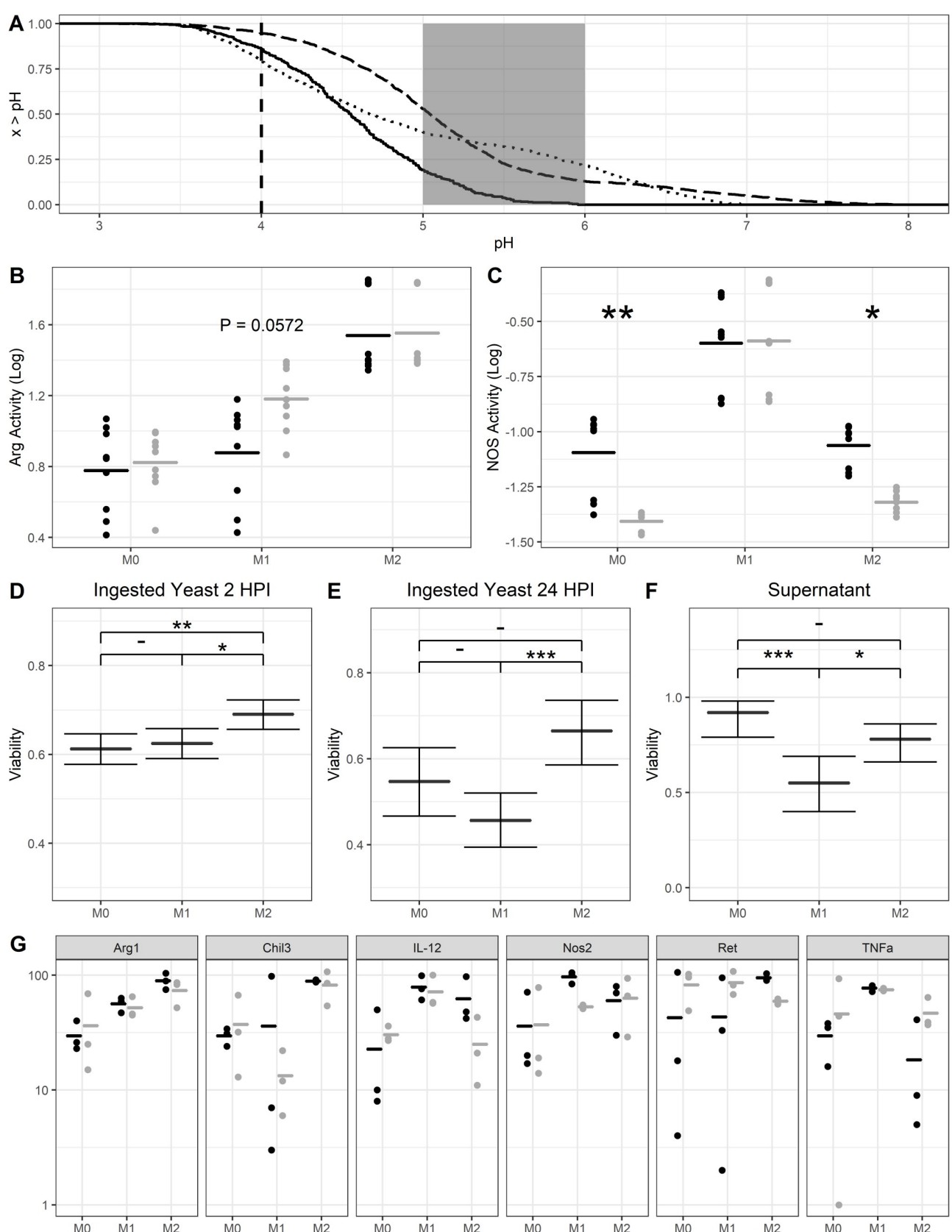

**Fig 1. pH and Oxidative response in BMDM populations. A.** Inverse empirical cumulative distribution functions for bead containing phagolysosome pH data measured in M0 (dotted), M1 (solid), and M2 (dashed) macrophage populations. The dashed line at pH 4 represents the point at which pH inhibits *C. neoformans* replication while the gray area between pH 5 and 6 represents the optimal growth pH for *C. neoformans*. Hospitality of each population is estimated by the number of phagolysosomes within each of these regions. Data was gathered from 3 independent replicates with n of 1490, 499, and 1419 for M0, M1, and M2 respectively. **B.** Arg-1 activity of differently polarized BMDMs infected (gray) or uninfected (black) with *C. neoformans*. M2 have the highest overall activity, and activity is promoted with infection in M1 populations. Data was gathered from 9 independent experiments with an n of 9 each. **C.** NOS activity of differently polarized BMDMs infected (gray) or uninfected (black) with *C. neoformans*. M1 have the highest overall activity, and activity is decreased with infection in both M0 and M2 populations. Significance determined by 2-way ANOVA with Tukey's HSD comparisons. Data was gathered from 9 independent experiments with an n of 9 each. **D.** Viability of *C. neoformans* ingested by macrophage populations after 2 h. Data was gathered from 3 independent experiments with n of 792, 821, and 785 for M0, M1, and M2 respectively. **E.** Viability of *C. neoformans* ingested by macrophage populations after 24 h. Data was gathered from 3 independent experiments with n of 159, 252, and 161 for M0, M1, and M2 respectively. **F.** Viability of extracellular *C. neoformans* after 24 h incubation with macrophages. Data was gathered from three independent experiments with n of 235, 237, and 257 for M0, M1, and M2 respectively. **G.** qPCR data from differently polarized BMDMs uninfected (black) and infected (gray) with *C. neoformans* 24 HPI. Values are normalized to the respective M0 uninfected fold induction and data was gathered from three independent replicates. *, **, *** represent $P < 0.05$, 0.01, and 0.001, respectively. ANOVA with Tukey comparisons were used in panels B and C while a test of equal proportions with Bonferroni correction was used in panels D and E. Boxplots signify median with 95% confidence interval tails.

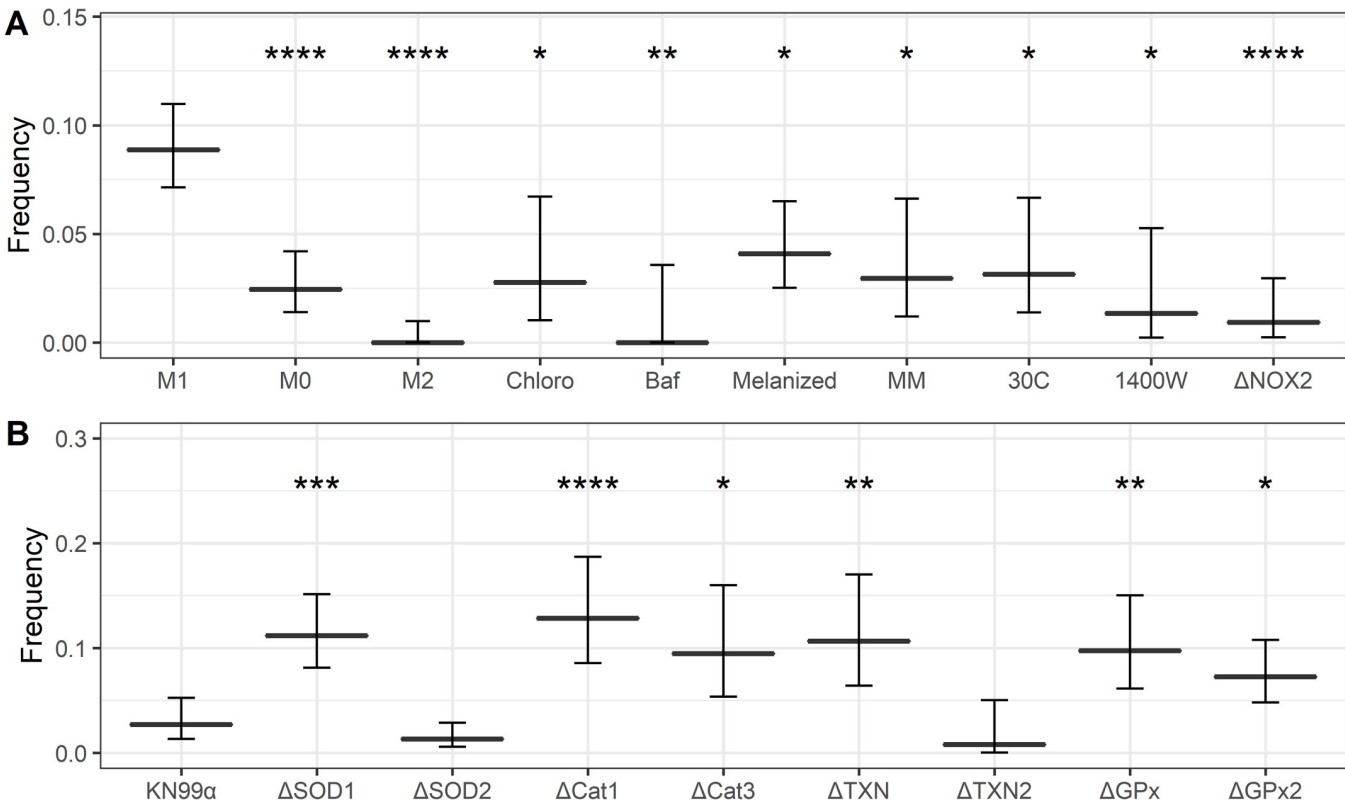

**Fig 2. Dragotcytosis frequencies of *C. neoformans* ingested BMDMs under various conditions. A.** *C. neoformans* strain H99 frequencies of Dragotcytosis. M0 and M2 macrophages have lower frequencies of Dragotcytosis overall compared to M1. Alkalizing the phagolysosomes of M1 macrophages with chloroquine (Chloro) or bafilomycin A1 (BafA), protecting from ROS with melanin (Melanized), stimulating capsule growth with minimal media conditions (MM), and inhibiting macrophage function with low temperature (30C) also abrogate Dragotcytosis frequency. Data was gathered from n of 901, 569, 478, 180, 130, 440, 203, 222, 149, and 619 for M1, M0, M2, Chloroquine (Chloro), Bafilomycin (Baf), Melanized, Capsule Induced (MM), 30C, 1400W, and ΔNOX2 respectively. **B.** *C. neoformans* strain KN99α Dragotcytosis frequencies among infected macrophages. Knocking out genes directly involved in oxidative stress mitigation increases frequency of Dragotcytosis. Data was gathered from n of 333, 340, 522, 187, 137, 150, 125, 195, and 330 for KN99α, ΔSOD1, ΔSOD2, ΔCat1, ΔCat3, ΔTXN, ΔTXN2, ΔGPx, and ΔGPx2 respectively. Graphs depict means with 95% confidence intervals. *, **, ***, **** denote $P < 0.05$, 0.01, 0.001, and 0.0001 via test of equal proportions compared to M1 with Bonferroni correction for multiple hypotheses. Boxplots signify median with 95% confidence interval tails.

## Macrophage polarization modulates oxidative response and affects *C. neoformans* viability

To determine whether the macrophage oxidative response varied between polarization states in a manner that would support our hypothesis on the triggers of Dragotcytosis frequency, we measured NOS and Arg1 activity between macrophage populations. M2 macrophages had the lowest NOS activity and the highest Arg1 activity, while M1 macrophages had the highest NOS activity with the lowest Arg1 activity. (**Fig 1**B and 1C). This data supports the notion that M2 macrophage phagolysosomes are more hospitable for *C. neoformans*. Additionally, by comparing enzymatic activity between macrophages infected or uninfected with *C. neoformans* we found that infection lowers NOS activity in M0 and M2 populations while increasing Arg1 activity in M1 populations (**Fig 1**B and 1C). Each of these changes would result in more hospitable phagolysosomes for *C. neoformans*.

We used GFP expression behind an actin promoter as a surrogate for viability for internalized *C. neoformans* in macrophages. Whether or not GFP negative cells are truly "dead" is a philosophical question beyond the scope of this paper, but a cell that does not express actin would certainly be incapable of division. Thus, we explored viability in terms of fungistatic rather than fungicidal qualities. We found that, at two hours post infection, *C. neoformans* ingested by M1 or M0 macrophages were less viable than those in M2 macrophages (**Fig 1**D). Additionally, we investigated the viability of extracellular *C. neoformans* and found that at 24 h post infection extracellular yeasts cultured with M1 macrophages are still inhibited (**Fig 1**E and 1F). This is consistent with prior reports that macrophages can inhibit extracellular *C. neoformans* cells [23,24]. We hypothesize this inhibition is due to the buildup of NO in the extracellular media. We found no extracellular crypto at 2 hours post infection consistent with our observations that exocytosis occurs hours after ingestion.

Finally, to investigate whether macrophage polarization was altered populationally during *C. neoformans* infection, we performed qPCR on RNA harvested from differently polarized macrophages 24 h after infection. We found that *C. neoformans* infection alone, in M0 macrophages, correlates with a populational skewing toward M2 polarization (**Fig 1**G). This effect is augmented in M2 macrophages activated beforehand with IL-4 suggesting that *C. neoformans* may actively skew macrophages toward the most hospitable state. We did not see such synergism when macrophages were stimulated with IFNγ and LPS, or when macrophages were not stimulated. It is unclear at this time how the polarization skewing is achieved. *C. neoformans* may actively encourage M2 macrophage polarization, possibly through secretion of intracellular proteins that affect macrophage function [25]. Alternatively, increased lysis frequency of M1 polarized macrophages could skew the population by simply removing M1 polarized macrophages (**S2 Fig**).

## Phagolysosome pH is associated with dragotcytosis frequency

To probe whether phagosomal pH modulated Dragotcytosis frequency rather than other downstream effects of polarization, we altered phagosomal pH in M1 macrophages with various drugs. Chloroquine is a weak base that localizes to phagolysosomes and can be used to alkalize macrophage phagolysosomes *in vitro*, and at 6 μM buffers phagolysosomes to a pH optimal *C. neoformans* growth. Bafilomycin A1 is a V-ATPase inhibitor which prevents protons from being pumped into the organelle. Alkalization of M1 cryptococcal phagosomes with either drug abrogated Dragotcytosis (**Fig 2**), consistent with the notion that phagolysosomal acidification triggers lateral cell transfer. Interestingly, treatment with sub-inhibitory concentrations of Fluconazole (20 μg / μL) and Amphotericin B (0.5 μg / μL) also reduced the frequency of Dragotcytosis. Unlike chloroquine, bafilomycin, and 1400W, which do not affect

the viability of fungal cells at the concentration used, Amphotericin B and Fluconazole both inhibit *C. neoformans*. The antifungal drug effects thus support our previous finding that only live *C. neoformans* are capable of initiating Dragotcytosis (**S5 Fig**).

## Disrupting macrophage anti-fungal function lowers dragotcytosis frequency

If Dragotcytosis is triggered by fungal cell stress, then further disrupting the macrophages' ability to inhibit *C. neoformans* would be expected to lower its overall frequency. We investigated this using two methodologies. First, we infected macrophages with *C. neoformans* induced to grow large capsules. The sheer size of cell as well as the anti-phagocytic properties of the capsule are known to inhibit macrophage uptake and promote fungal survival. Furthermore, the *C. neoformans* capsule was previously shown to have buffering potential and raise the phagolysosomal pH, and consistent with that observation we observed that cryptococcal cells with induced larger capsules had higher phagolysosomal pH than those with smaller capsules (**S6 Fig**) and acapsular *C. neoformans* resulting in lower phagolysosomal pH [10]. We found that capsule induced *C. neoformans* Dragotcytosed less frequently and Vomocytosed dramatically more often compared to wild-type, resulting in no overall change when comparing both types of events together (**Fig 2**A). Next, we infected macrophages with wild-type *C. neoformans* at 30 $^0$C, a temperature optimal for the fungus and suboptimal for the macrophage. We again found that Dragotcytosis was abrogated (**Fig 2**A). Taken together these results support the hypothesis that Dragotcytosis is induced when *C. neoformans* are stressed or overwhelmed by host macrophage defenses. Finally, we investigated whether melanized *C. neoformans* Dragotcytosed at different frequencies as melanin is known to protect from phagolysosomal stressors [26–28]. We found that melanized yeast underwent Dragotcytosis at a reduced rate (**Fig 2**A). We confirmed this finding with two alternative methods: using NOX2 knockout mice that cannot make NO and by inhibiting oxidative burst with 1400W. In both cases, Dragotcytosis frequencies were reduced (**Fig 2**A). Taken together, these data suggest that Dragotcytosis could be triggered by general cellular stress within the phagolysosome rather than pH specifically.

## Oxidative stress modulates dragotcytosis frequency

To further probe whether oxidative stress had a role in triggering host cell escape, we investigated several mutant *C. neoformans* strains deficient in various oxidative stress mitigation pathways including: Superoxide Dismutases 1 and 2 (SOD1, SOD2), Catalases 1 and 3 (Cat1, Cat3), Thiorexodin 1 (TXN), and Glutathione Peroxidases 1 and 2 (GPx and GPx2, 19). We first observed that the mutant library parental strain KN99α undergoes Dragotcytosis at a significantly reduced rate compared to strain H99 ($P < 0.0001$ via test of equal proportions, **Fig 2**). ΔSOD1, ΔCat1, ΔCat3, ΔTXN, ΔGPx, and ΔGPx2 strains of *C. neoformans* underwent Dragotcytosis at an increased frequency compared to the parental strain KN99α, while Dragotcytosis frequency in ΔSOD2 and ΔTXN2 remained unchanged (**Figs 2B and S7**). Furthermore, both supplementing an M1 skewed macrophage population with 1400W prior to infection with ΔSOD1 and infecting NOX2 knockout mice completely abrogated the increased Dragotcytosis phenotype (**S8 Fig**). Overall, these data support the notion that Dragotcytosis is triggered by a stressed yeast, and that increasing stress on the yeast results in increased Dragotcytosis.

To investigate whether the frequency of Dragotcytosis is increased significantly enough in these knockout mutant strains (ΔSDO1, ΔSOD2, ΔCat1, ΔCat3, ΔTXN, ΔTXN2, ΔGPx, and ΔGPx2) to completely offset their increased susceptibility to ROS, we measured total cell

counts and cell viability of intracellular and extracellular yeast during BMDM infection. We found that the total cell population was unchanged, but the mutant strains suffered reduced viability compared to wild type (**S9 Fig**).

### Time to host exit initiation is stochastic for exocytosis events

If Vomocytosis and/or Dragotcytosis are triggered by hostile pH and final pH is determined stochastically from a normal distribution [10], then these host cell exocytosis events should also display stochastic dynamics resembling those of the upstream pH trigger. To explore whether the stochasticity of phagolysosome acidification is preserved in this system, we measured the time at which each process occurs throughout a series of videos of macrophages infected with *C. neoformans*. We found that while the time of exocytosis events was not normally distributed, these events were triggered stochastically with no forbidden ordinal patterns observed (**Fig 3A and 3B**). The absence of forbidden ordinals implies that the process is not deterministic, ruling out chaotic dynamics.

### Simulation data suggests dragotcytosis at low ph benefits cryptococcal cells

We hypothesized that triggering Dragotcytosis, or non-lytic exocytosis, in phagolysosomes of low pH would benefit *C. neoformans* cells by lowering the total number of yeasts within inhibitory phagolysosomes. The argument for non-lytic exocytosis is intuitive, in that the yeast leaves the phagolysosome altogether and could only result in fewer inhibitory phagolysosomes. Dragotcytosis however is more nuanced as there is no obvious way for *C. neoformans* to know whether the new phagolysosome will be more or less hostile to the yeast. To explore this hypothesis, we modeled phagolysosomal pH distributions based on observed pH measurements of bead containing BMDM phagolysosomes and the established bet-hedging strategy of host macrophages in silico. We found that if a *C. neoformans* yeast were to trigger Dragotcytosis even just a single time if their original phagolysosome dropped to pH < 4, it would be enough to drop the proportion of inhibitory phagosomes from ~15% to ~1% (**Fig 3**C and 3D). The actual effect on *C. neoformans* survival would be even more pronounced since Dragotcytosis events are not limited to one event per yeast in vitro/vivo. We then investigated how beneficial Dragotcytosis would be at a range of pH cutoffs, assuming that an ingested *C. neoformans* would be equally likely to migrate into an M0, M1, or M2 macrophage. In the context of pH, the benefit of a Dragotcytosis event increases steadily the further away from an optimal pH the original phagolysosome is (**Fig 3**E).

### *Cryptococcus neoformans* does not display an inherent bias for destination macrophages

While there is no obvious and apparent method for *C. neoformans* to specifically target M2 macrophages as the Dragotcytosis destination, such a system could still be possible. The donor and acceptor macrophages maintain physical contact during the Dragotcytosis process, and an immune synapse could transduce signals detectable by the ingested yeast. To investigate whether *C. neoformans* shows a preference during Dragotcytosis we compared the rates of transfer from infected M1 macrophages to uninfected M1 or M2 macrophages. If there is a preference, then we would expect to observe an increased amount of *C. neoformans* residing in M2 macrophages 24 HPI; if there were no preference, we would expect to see equal transfer into M1 and M2 macrophages. Instead, we see roughly an equivalent amount of transfer into M1 and M2 macrophages with slightly less transfer into M0 macrophages (**Fig 4**A and 4B). Based on a model of Dragotcytosis dynamics, we expect it would take approximately 4 to 5

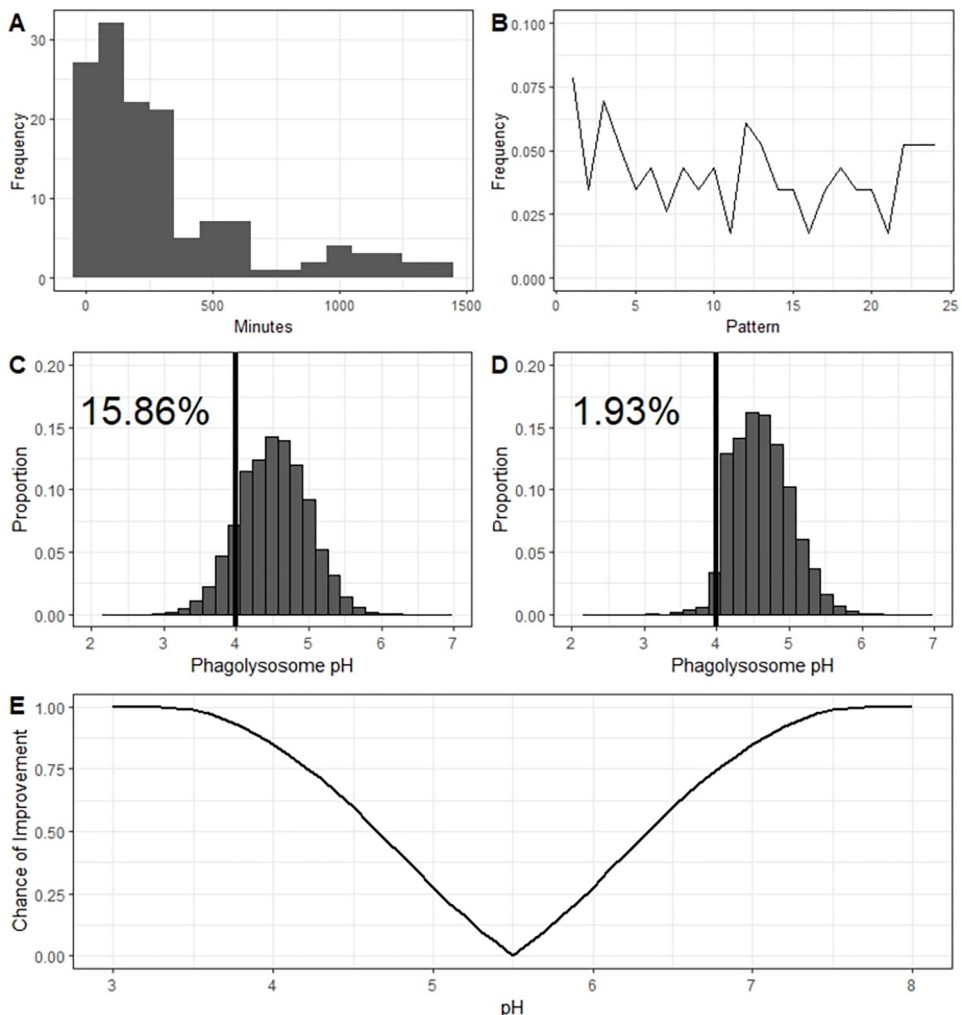

**Fig 3. Dynamics of Dragotcytosis. A.** Distribution of the times at which *C. neoformans* yeasts initiated host cell exit strategies. Both Vomocytosis and Dragotcytosis events are represented here. Bin widths are set to 100 and the data depicted spans 139 samples from 12 independent experiments. **B.** Ordinal pattern analysis for the intervals between events. Intervals were gathered and analyzed within experiments and total proportions of individual ordinal patterns summed between experiments. **C.** Hypothetical phagolysosomal pH distributions if *C. neoformans* particles were to undergo no Dragotcytosis events. **D.** Modelled phagolysosomal pH distributions if *C. neoformans* particles were to undergo one Dragotcytosis event if their resident phagolysosomal pH is < 4. Even a single round of low pH triggered Dragotcytosis drastically shifts the distribution to the right resulting in a greater proportion of *C. neoformans* hospitable phagolysosomes. Labelled percent values reflect the proportion of hypothetical phagolysosomes inhibitory to *C. neoformans* (pH < 4, black vertical line) in each scenario. **E.** Hypothetical chance of an ingested *C. neoformans* to find a more hospitable phagolysosome in the context of pH assuming equal likelihood of migrating to an M0, M1, or M2 macrophage.

sequential transfer events before we would be able to detect differences in population based on random chance alone (**Fig 4**C).

## Proposed model for triggering dragotcytosis

Dragotcytosis requires live *C. neoformans* and its frequency is modulated by variables that effect stress on the fungal cell (Table 2). We synthesize these observations into a model whereby stress in the phagolysosome, resulting in fungal cell damage, triggers a program for

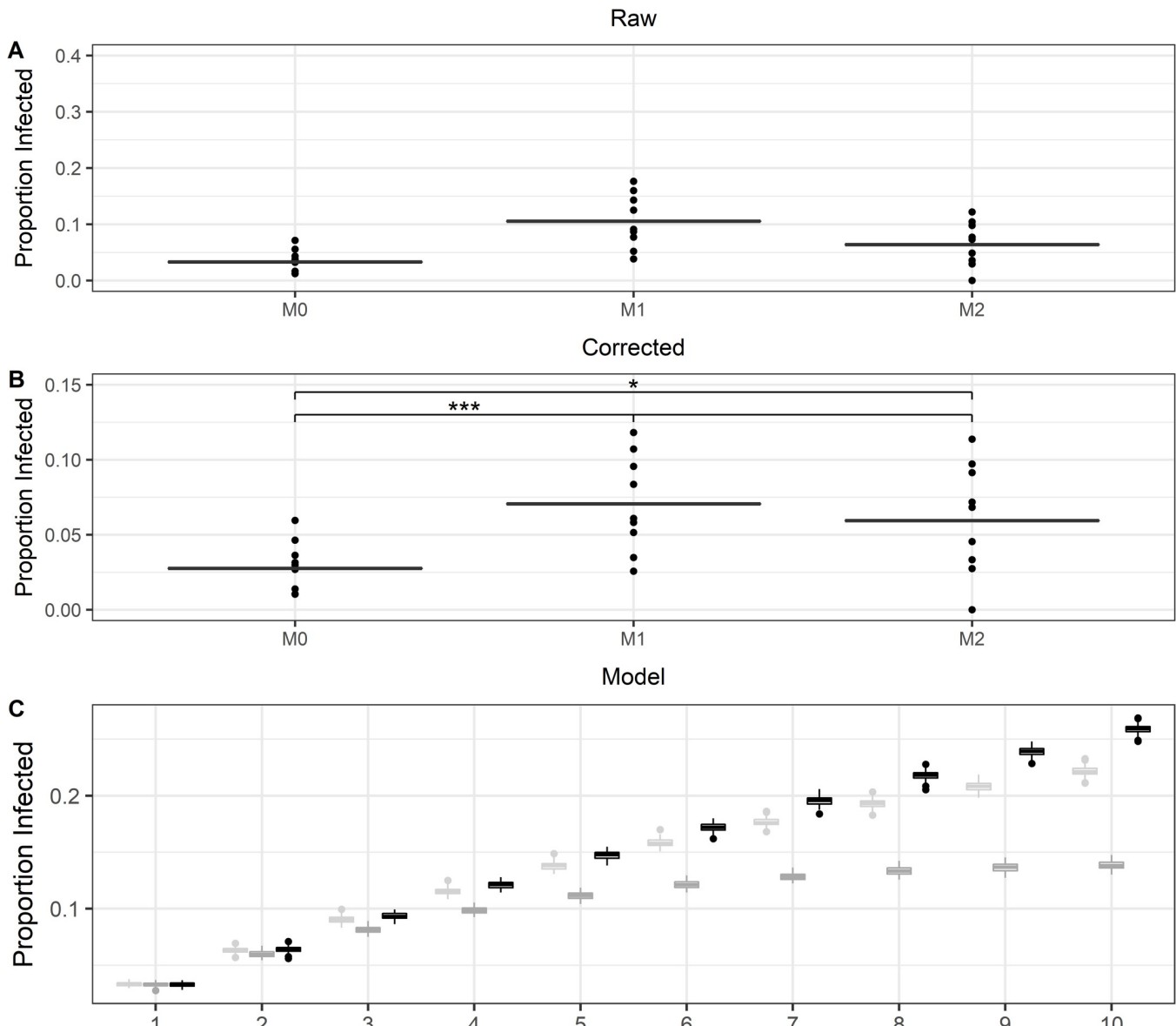

**Fig 4. Comparisons of labelled macrophage populations and the proportion of which have ingested *C. neoformans* 24 hours post infection. A.** The uncorrected proportion of adherent macrophages containing *C. neoformans* at the end of the infection. Data was collected from 10 independent experiments. **B.** The proportion of macrophages containing yeast after correcting for floating cells. Both the proportion of labelled cells compared to the original population of infected M1 macrophages as well as the frequency of labelled cells which ingested *C. neoformans* remained consistent throughout the time course. *, *** signify $P < 0.05$ and 0.001 via ANOVA with Tukey multiple comparisons. **C.** A model depicting expected proportions of infected cells based only on random chance after sequential Dragotcytosis events for M0 (light gray), M1 (dark gray), and M2 (black) macrophages. Boxplots signify median with 95% confidence interval tails.

exiting the host cell. We know that exocytosis is associated with actin flashes [7,29] and posit that stressed cells activate a program that manipulates the host cell to promote a choreography of events resulting in transit from the phagolysosome to outside of the host cell (Fig 5). If these events lead to exocytosis at the cell membrane region in contact with an adjoining macrophage, then non-lytic exocytosis becomes Dragotcytosis. Given that non-lytic exocytosis has been observed in mammalian, fish, insect, and protozoal cells the exit program is likely to target conserved functions in eukaryotic phagocytic cells [6,8,30–32]. As to whether apposition of

**Table 2. Summary of variable effects on Dragotcytosis frequency.**

| Frequency | Variable | Effect |
|---|---|---|
| Increase | Macrophage M1 Polarization | Enhanced antifungal activity (increased NO) |
| | Decreased Phagosome pH | Increased phagolysosome fungistatic mechanisms Reduced fungal growth |
| | CN Superoxide 1 Deficiency | Enhanced fungal susceptibility to oxidants |
| | CN Catalase 1 and 3 Deficiency | Enhanced fungal susceptibility to oxidants |
| | CN Thioredoxin Deficiency | Enhanced fungal susceptibility to oxidants |
| | CN Glutathione Peroxidase Deficiency | Enhanced fungal susceptibility to oxidants |
| Decrease | Macrophage M2 Polarization | Reduced antifungal activity |
| | Increased Phagosome pH | Reduced phagolysosome fungistatic mechanisms Enhanced fungal replication |
| | Bafilomycin A1 and Chloroquine Treatment | Increases phagolysosomal pH, reduced phagolysosome fungistatic mechanisms |
| | Fluconazole and Amphotericin B Treatment | Disrupts fungal ergosterol processes |
| | NOX2 KO or 1400W Treatment | Absent or reduced NOS antimicrobial activity |
| | Melanin | Reduced fungal susceptibility to oxidants |
| | CN Urease Deficiency | Decreased phagosomal pH, increased intracellular CN replication, disrupted phagolysosomal membrane, increased apoptotic macrophages [5] |
| | Increased Capsule Size | Increased buffering potential, reduced macrophage phagocytosis efficiency |
| | Temperature (30 $^0$C) | Reduced macrophage function |

two macrophages induces changes in the infected cell that are sensed by the fungal cell is unknown and a subject for future investigation.

## Discussion

Dragotcytosis is a cellular process by which *C. neoformans* yeasts within a host macrophage phagolysosome transfer to another, proximal macrophage without lysis of the initial host [6,8,9,13]. During recent investigations into the receptors used in Dragotcytosis and into the phagolysosomal acidification dynamics of macrophages, we found a correlation between phagolysosomal hospitality and the frequency of Dragotcytosis, suggesting that Dragotcytosis is beneficial to, and triggered by, ingested *C. neoformans* [9,10]. Arguably, Dragotcytosis also benefits the macrophage by relieving it of a fungal cell that causes cellular damage, although the net benefit versus costs of this transfer event for the microbe and host cell remains uncertain. The fact that Dragotcytosis was initiated by the fungal cell led us to hypothesize that fungal discomfort in the phagosome was a trigger for cell-to-cell transfer and in this study, we demonstrate that cellular stress on *C. neoformans* leads to increased Dragotcytosis frequency.

Previous studies showed that M2 macrophages and Th2 skewed immune responses are more permissive to *C. neoformans* infection [33–36]. Our initial observation that M2 macrophages have, on average, higher pH phagolysosomes compared to M1 macrophages when containing inert particles offers a partial explanation of why these macrophages are more hospitable. Specifically, M2 macrophage populations had the fewest phagolysosomes at an inhibitory pH (pH < 4) and the most phagolysosomes in an optimal pH range (5 < pH < 6) for fungal growth [22]. Additionally, we noticed that *C. neoformans* underwent fewer

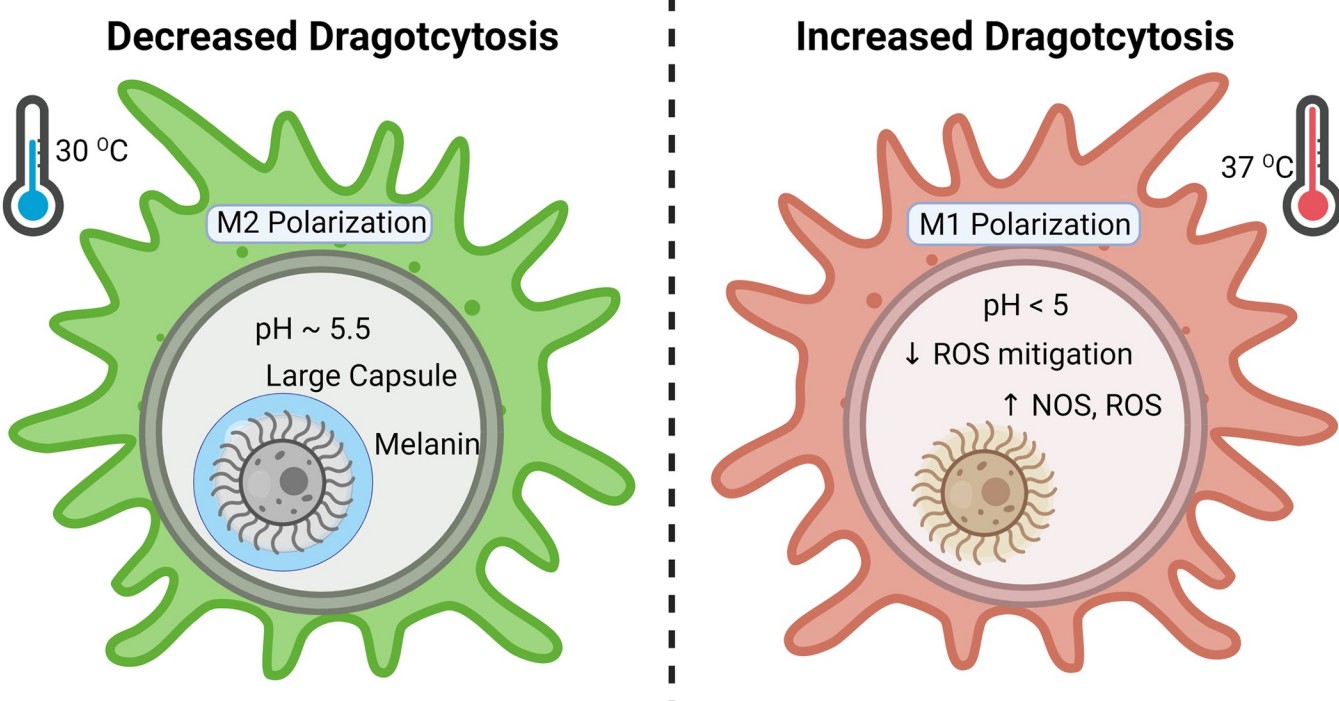

**Fig 5. Graphical summary of conditions with effects on Dragotcytosis frequency.** Figure was created with BioRender.com.

Dragotcytosis events when residing in M2 macrophage phagolysosomes compared to yeast cells resident in M1 macrophages. From this correlation, we hypothesized that if the phagolysosomal pH dropped too low for an ingested *C. neoformans* to counter acidity with its polysaccharide capsule or urease activity [3,5,37,38], then it triggered either non-lytic exocytosis or Dragotcytosis as an escape mechanism that would bring the yeast cell to the less acidic extracellular environment or into a phagolysosome of more likely to have a hospitable pH, respectively.

Since phagolysosomal pH is stochastically determined with a distribution in which most phagosomes have pH > 4, it is likely that *C. neoformans* residing in a phagosome with pH < 4 would find a more hospitable home in another macrophage after a Dragotcytosis event [10]. Consequently, we probed how Dragotcytosis could modulate the effectiveness of the macrophage acidification bet hedging strategy via in silico modeling and found that even a single Dragotcytosis event triggered for *C. neoformans* residing in low pH phagolysosomes would significantly increase the likelihood of finding a less acidic phagosome. These findings could also help explain how *C. neoformans* replicates faster inside macrophages *in vivo* and *in vitro* than in the extracellular space, which has been attributed to faster replication in the acidic pH of the average phagosome [39–41]. Despite not all phagosomes being conducive to faster fungal growth, sequential Dragotcytosis events could lead *C. neoformans* to find phagolysosomes more permissive to rapid growth during an infection. These results fit well with the known data of *C. neoformans* infection in human macrophages. Human macrophage phagolysosomes acidify with different dynamics than those of mice. For example, human M2 macrophage phagolysosomes acidify to a lower pH than mouse macrophage phagolysosomes [42] and, perhaps as a result, humans are markedly resistant to *C. neoformans* infection which poses little threat to most immune competent hosts.

To probe deeper into whether there was a causal association between average phagolysosomal pH of M1 macrophages and the frequency of Dragotcytosis, we treated macrophages with either chloroquine or bafilomycin A1, which increased phagolysosomal pH near the optimal growth range of *C. neoformans* and comparable to the observed average phagolysosomal pH of M2 macrophage populations. We found that both chloroquine and bafilomycin A1 drastically reduced Dragotcytosis events, supporting the hypothesis that cellular stress is the stimulus for Dragotcytosis. We are confident that the drug treatments themselves did not have a direct effect on *C. neoformans* growth, virulence, or survivability at the concentrations used but rather modulated the host macrophage response. Chloroquine does not significantly reduce *C. neoformans* growth or viability unless ingested by macrophages and at concentrations higher than used in these experiments [43]. While high concentrations Bafilomycin A1 has antifungal [44] and even inhibits *C. neoformans* melanization [45], the concentrations used here are much lower than those associated with antifungal effects [46]. We also treated cells with sub-inhibitory concentrations of fluconazole and amphotericin B, to explore whether these drugs would also stress the yeast in a way that promotes Dragotcytosis [47]. Interestingly, both anti-fungal drugs reduced the overall frequency of Dragotcytosis possibly because the concentrations were still too disruptive to fungal metabolism, as *C. neoformans* growth is inhibited by both drugs, and the yeast was too compromised to transfer. In this regard, both drugs affect fungal membrane ergosterol integrity and fungal lipids may be required for the Dragotcytosis process. Alternatively, ergosterol disruption may not be sensed by the fungal machinery that triggers Dragotcytosis.

To further investigate the oxidative trigger hypothesis, we characterized Dragotcytosis frequencies of several *C. neoformans* strain KN99α mutants from a mutant knockout library during infection. We first observed that KN99α undergoes Dragotcytosis at a reduced rate compared to H99. We do not know the specific genetic or phenotypic changes responsible for the baseline reduced Dragotcytosis rate but it is not unexpected as KN99α is considered a hypervirulent strain due to multiple factors [48–50]. We selected a variety of potential candidates based on known pathways relevant to our cellular stress hypothesis. We found that SOD1, Cat1, Cat3, TXN, GPx, and GPx2 deficient mutants underwent Dragotcytosis much more frequently than the wild-type parental strain, supporting the notion that Dragotcytosis is triggered by the accumulation of oxidative stress. Interestingly, the ΔSOD2 and ΔTXN2 mutants did not exhibit increased frequency of Dragotcytosis which may reflect the nature of the phagolysosomal oxidative stress, as SOD1 and SOD2 are known to have different efficacy against different oxidative stresses [51].

If oxidative stress triggers Dragotcytosis, then reducing stress within the phagolysosome should reduce the frequency of Dragotcytosis. Melanin is known to protect *C. neoformans* from oxidative stress [52]. Melanized *C. neoformans* triggered Dragotcytosis at a lower frequency compared to non-melanized *C. neoformans*. Similarly, fewer Dragotcytosis events were observed in infections of NOX2 knockout mice incapable of oxidative burst and in wild-type macrophages treated with the NOS inhibitor 1400W. These conditions also restored Dragotcytosis frequency in the ΔSOD1 strain to the same frequency as wild-type KN99α. Finally, the cryptococcal capsule is known to protect against both pH and ROS in macrophages. Macrophages infected with *C. neoformans* with large capsules manifested Dragotcytosis at a reduced rate, though Vomocytosis occurred at an increased rate. It is less clear whether Vomocytosis is triggered by the yeast or host as heat killed *C. neoformans* have been observed to undergo Vomocytosis, albeit at a drastically reduced rate [7], raising the question of what could trigger one pathway opposed to the other. When macrophage infection was done at a temperature optimal for *C. neoformans* and suboptimal for macrophages: 30 $^0$C we again found that Dragotcytosis frequency was reduced.

Following our observations that increasingly stressful conditions increased Dragotcytosis, and that increased Dragotcytosis is associated with *C. neoformans* survival during these *in vitro* infections, we investigated whether this egress from hostile phagolysosomes would offset the increased ROS damage sustained in the ROS knockout mutant library. Thus, we measured intracellular and extracellular total cell counts and cell viability of all ROS knockout mutants used in the study. We found that the overall number of yeasts did not significantly change after 24 h infection, but the viability of each ROS mutant was significantly decreased compared to the parental strain after ingestion. Extracellular yeasts were more variable in their viability with ΔSOD2 and ΔGPx suffering no loss in viability.

Next, we considered whether *C. neoformans* induced a bias toward M2 macrophages as acceptor macrophages in the Dragotcytosis process. Logically it would benefit the yeast as M2 macrophages were found to be more permissive, but a mechanism by which the yeast could sense interaction with an M2 macrophage and initiate the Dragotcytosis process all from within the phagolysosome is difficult to imagine. Thus, we designed an experiment in which infected M1 macrophages would be seeded with labelled and uninfected M1 or M2 macrophages and we would compare the rate of transfer. If there was a preference, we should see an unequal rate of transfer between the two populations. Instead, we observed equal rates of transfer into both macrophage types. We expect that, given enough time, the distribution would shift to favor M2 macrophages by random chance. However, this was not observed due to the infrequency of sequential macrophage-to-macrophage transfers by a single cryptococcal cell during the first 24 hours of experimental observation. Thus, we find it more likely that *C. neoformans* initiates a transfer without prior knowledge of the polarization state of the receiver macrophage and instead continues to transfer until a hospitable phagolysosome is discovered by chance.

Dragotcytosis was stochastically initiated, reflecting the dynamics of phagolysosome acidification [10]. However, unlike phagosomal acidification, which manifested a normal distribution, the distribution of Dragotcytosis as a function of time was skewed away from normality. Therefore, while phagolysosomal pH contributed to triggering Dragotcytosis, it is likely not the only important trigger, a hypothesis supported by previous observations that inhibiting pH buffering via urease mutation did not increase Dragotcytosis events as would be expected if the process was triggered by pH alone [5]. Instead, Dragotcytosis may be triggered by an accumulation of diverse sources of cellular stress encountered in the phagolysosome. Whereas the endpoint of phagolysosomal acidification represents a bet hedging strategy by macrophages to control ingested microbes [10], *C. neoformans* also appears to be employing a bet hedging strategy in exocytosis that is triggered by the bet-hedging strategy in macrophage phagolysosome acidification. In fact, *C. neoformans* may engage in its own bet hedging strategy wherein Dragotcytosis is triggered past a certain threshold of stress in which it becomes more likely that the ingested yeast will migrate to a more hospitable phagolysosome. This results in the interesting situation whereby a defensive strategy by the host cells is co-opted for survival by the fungal cell. Hence, chance outcomes in the phagolysosome trigger events that produce chance outcomes in macrophage exocytosis, implying connectivity between these two cellular processes with regards to their system dynamics. Stochastic processes are random and thus inherently unpredictable, suggesting a fundamental unpredictability to cellular processes that could extend to making host-microbe interactions not predictable.

We previously proposed that Dragotcytosis was the result of sequential Vomocytosis and phagocytosis events on cell adjacent macrophages [9]. If this is the case, then the frequency of Vomocytosis events should correlate with that of Dragotcytosis events as the frequency of any non-lytic escape is determined by overall cellular stress but the decision to undergo Dragotcytosis as opposed to Vomocytosis would rely on additional, more complex, parameters.

Consistent with this thesis, we observed a correlation between Dragotcytosis and Vomocytosis frequencies.

In summary, our results consistently show that conditions that increase or decrease *C. neoformans* stress are associated with enhanced and reduced Dragotcytosis, respectively. ROS stress appears to be a particularly potent regulator of Dragotcytosis, but it may be impossible to narrow down Dragotcytosis triggers to one specific stressor. Drugs that inhibit phagosome acidification can also inhibit autophagy, and vice versa, making it difficult to parse out the effects of only one of these systems at a time. Even the generation of reactive oxygen species is innately tied to these processes, and phagolysosomal pH itself, with the concentration of ROS in the phagolysosome increasing alongside pH with previously measured concentrations of 50 μM $O_2^{\bullet-}$ at pH 7.4 and 2 μM at pH 4.5 [15]. We find it more likely that a combination of cellular stresses triggers Dragotcytosis. We note how the normally distributed phagolysosomal pH following phagocytosis randomly includes some phagolysosomes that are inhospitable to fungal cells which, in turn, triggers exocytosis with its own stochastic dynamics with a different distribution from normally distributed phagolysosome pHs [10], as other factors contribute to the exit phenomenon and skew the distribution away from normality. Hence, stochastic dynamics in phagosomal acidification beget stochastic dynamics in non-lytic exocytosis, implying a fundamentally unpredictable host-microbe interaction at the level of cellular organelles. Such unpredictability at the cellular level is a likely contributor to our inability to predict the outcome of host-microbe interactions at the organismal level [53].

## Supporting information

**S1 Fig. Frequency of Dragotcytosis events of ingested *C. neoformans* H99 yeasts according to population size of infected or total macrophages. A.** Dragotcytosis frequency did not correlate with total number of infected macrophages. **B.** Dragotcytosis frequency did not correlate with the density of total macrophages. **C.** Dragotcytosis frequency did not correlate with the proportion of uninfected to infected macrophages. Linear regressions were performed on each dataset with 95% confidence intervals (gray). **D.** Frequencies of Vomocytosis and Dragotcytosis obtained from 22 independent movies of *C. neoformans* strain H99 infection of macrophages. The Pearson correlation between the two processes is 0.41 with $P = 0.055$ and the Spearman correlation is 0.48 with $P = 0.025$.
(TIFF)

**S2 Fig. Event frequencies of *C. neoformans* strain H99 ingested by BMDMs under various pH related conditions. A.** Vomocytosis frequency among infected macrophages. **B.** Combined frequency of Dragotcytosis and Vomocytosis among infected macrophages. **C.** Lysis frequency among infected macrophages. All conditions have similar frequency. Event frequencies of wild-type KN99α strain and mutant *C. neoformans* after ingestion by M1 polarized BMDMs. \*, \*\*, \*\*\*, \*\*\*\* signify $P < 0.05, 0.01, 0.001$, and 0.0001 via test of equal proportions, respectively. Bonferroni correction was applied for multiple hypotheses. Boxplots signify median with 95% confidence interval tails. Data was gathered from n of 901, 569, 478, 180, 130, 440, 203, 222, 149, and 619 for M1, M0, M2, Chloroquine (Chloro), Bafilomycin (Baf), Melanized, Capsule Induced (MM), 30C, 1400W, and ΔNOX2 respectively.
(TIFF)

**S3 Fig. Capsule and cell body size of *C. neoformans* isolated 24 hours after ingestion by BMDMs.** Capsules and cell bodies are measured by preparing and imaging India Ink slides and a previously published[20] measuring code. No significant differences were found between the polarization states of host macrophages. Boxplots signify median with 95%

confidence interval tails. For each experiment, data was collected for at least three independent experiments and total n of 563, 870, and 554 for M0, M1, and M2 respectively.
(TIFF)

**S4 Fig. Fluorescence ratio of ingested particles as a measurement of phagolysosomal pH compared to total volume of ingested particles. A.** Fluorescence ratio of ingested *C. neoformans*. We found no significant correlation. **B.** Fluorescence ratio of ingested inert latex beads of 0.6 μm diameter. Even with a cluster of four particles within a single phagosome we did not detect a threshold at which size alone disrupts the phagolysosomal pH. Boxplots signify median with 95% confidence interval tails and n of 16, 30, 18, 16 for cluster sizes 1, 2, 3, and 4 respectively.
(TIFF)

**S5 Fig. Growth curves of *C. neoformans* strain H99 incubated with various drugs from three biological replicates.** Cultures were seeded at $10^4$ cells / mL and grown for 72 h at 30˚C with 120 rpm shaking. Fluconazole and Amphotericin B inhibit cryptococcal growth while Bafilomycin, Chloroquine, and 1400W do not. Lines with shaded areas represent 95% CI of 3 independent replicates.
(JPEG)

**S6 Fig. Observed phagolysosomal pH of BMDMs infected with *C. neoformans* cultured in media rich capsule non-inducing (YPD) or glucose scarce capsule inducing (MM) conditions.** Data was collected from three independent experiments and total n = 83 and 63 phagolysosomes, respectively. **** signifies $P < 0.0001$ via two tailed t-test. Boxplots signify median with 95% confidence interval tails.
(TIFF)

**S7 Fig. Event frequencies of *C. neoformans* strain KN99α and knockout mutants ingested by M1 BMDMs. A.** Vomocytosis frequency among infected macrophages **B.** Combined Dragotcytosis and Vomocytosis frequency among infected macrophages. **C.** Lysis frequency among infected macrophages. Graphs depict means with 95% confidence intervals. *, **, ***, **** signify $P < 0.05, 0.01, 0.001,$ and $0.0001$ via test of equal proportions, respectively. Bonferroni correction was applied for multiple hypotheses. Boxplots signify median with 95% confidence interval tails. Data was gathered from n of 333, 340, 522, 187, 137, 150, 125, 195, and 330 for KN99α, ΔSOD1, ΔSOD2, ΔCat1, ΔCat3, ΔTXN, ΔTXN2, ΔGPx, and ΔGPx2 respectively.
(TIFF)

**S8 Fig. Dragotcytosis frequency of ΔSOD1 *C. neoformans* mutants ingested by macrophages treated with iNOS inhibitor 1400W or from NOX2 knockout mice compared to the parental strain and untreated ΔSOD1.** Both 1400W treatment and ΔNOX2 macrophage infection eliminate the increased Dragotcytosis phenotype, returning Dragotcytosis frequency to wild-type level. ** and **** represent $P < 0.01$ and $0.0001$ via test of equal proportions with Bonferroni multiple hypothesis correction. Boxplots signify median with 95% confidence interval tails. For each experiment, data was collected for at least three independent experiments and with n of 333, 340, 136, and 158 infected macrophages for KN99α, ΔSOD1, ΔSOD1-1400W, and ΔSOD1-ΔNOX2, respectively.
(TIFF)

**S9 Fig. Total cell and viability counts of wild-type and ROS mutant strains after a 24 h infection in BMDMs. A.** Total yeast cells recovered intracellularly normalized to the number of recovered wild-type cells. **B.** Total yeast cells recovered extracellularly normalized to the number of recovered wild-type cells. Total cell counts were measured via hemocytometer

count and no significant deviation was detected by two-tailed t-tests with Bonferroni correction. Samples were measured from 3–4 independent experiments. **C.** Viability of yeast recovered intracellularly. **D.** Viability of yeast recovered extracellularly. Viability was measured via trypan blue stain and compared to wild type via test of equal proportions with Bonferroni correction. Samples were taken from three independent experiments with n of 2029, 1580, 1555, 1548, 2485, 1454, 2028, 1426, and 2583 yeast cells for KN99$\alpha$, $\Delta$SOD1, $\Delta$SOD2, $\Delta$Cat1, $\Delta$Cat3, $\Delta$TXN, $\Delta$TXN2, $\Delta$GPx, and $\Delta$GPx2 respectively. $^*$ and $^{****}$ denote $P < 0.05$ and 0.0001, respectively.
(TIFF)

**S10 Fig. Examples of microscopy images of infected BMDMs and *C. neoformans* host escape events. A.** Examples of each type of host escape event taken from time lapse microscopy. The yeast undergoing each event is outlined in the first frame in green and each frame is 2 min apart. **B.** Representative images of each polarization type infected with *C. neoformans*. Scale bar represents 20 μm and is consistent across all images.
(TIF)

## Author Contributions

**Conceptualization:** Quigly Dragotakes, Arturo Casadevall.

**Data curation:** Quigly Dragotakes.

**Formal analysis:** Quigly Dragotakes.

**Funding acquisition:** Arturo Casadevall.

**Investigation:** Quigly Dragotakes, Ella Jacobs, Lia Sanchez Ramirez, Olivia Insun Yoon, Caitlin Perez-Stable, Hope Eden, Jenlu Pagnotta, Raghav Vij.

**Methodology:** Quigly Dragotakes, Ella Jacobs, Aviv Bergman, Franco D'Alessio, Arturo Casadevall.

**Project administration:** Quigly Dragotakes, Arturo Casadevall.

**Resources:** Aviv Bergman, Franco D'Alessio, Arturo Casadevall.

**Supervision:** Aviv Bergman, Franco D'Alessio, Arturo Casadevall.

**Validation:** Quigly Dragotakes, Aviv Bergman, Franco D'Alessio, Arturo Casadevall.

**Visualization:** Quigly Dragotakes, Ella Jacobs, Jenlu Pagnotta, Raghav Vij.

**Writing – original draft:** Quigly Dragotakes, Arturo Casadevall.

**Writing – review & editing:** Quigly Dragotakes, Arturo Casadevall.

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
