## [Decision Letter · Decision Letter 0]

23 Jun 2022

Dear Mr. Dragotakes,

We are pleased to inform you that your manuscript 'Bet-hedging antimicrobial strategies in macrophage phagosome acidification drive the dynamics of Cryptococcus neoformans intracellular escape mechanisms' has been provisionally accepted for publication in PLOS Pathogens.

Best regards,

Simon A. Johnston

Guest Editor

PLOS Pathogens

Xiaorong Lin

Section Editor

PLOS Pathogens

Kasturi Haldar

Editor-in-Chief

PLOS Pathogens

orcid.org/0000-0001-5065-158X

Michael Malim

Editor-in-Chief

PLOS Pathogens

orcid.org/0000-0002-7699-2064

Reviewer Comments (if any, and for reference):

Reviewer's Responses to Questions

**Part I - Summary**

Reviewer #1: See first review.

I am satisfied with the arguments offered in the rebuttal letter and with the changes introduced as additional experiments and text modifications.

Reviewer #2: Reviews were done accordingly

**Part II – Major Issues: Key Experiments Required for Acceptance**

Reviewer #1: (No Response)

Reviewer #2: Reviews were done accordingly

**Part III – Minor Issues: Editorial and Data Presentation Modifications**

Reviewer #1: (No Response)

Reviewer #2: Reviews were done accordingly

PLOS authors have the option to publish the peer review history of their article (what does this mean?). If published, this will include your full peer review and any attached files.

Reviewer #1: No

Reviewer #2: No

Figure Files:

Data Requirements:

Reproducibility:

References:

---

## [Editor Report · Acceptance letter]

7 Jul 2022

Dear Dr. Casadevall,

We are delighted to inform you that your manuscript, "Bet-hedging antimicrobial strategies in macrophage phagosome acidification drive the dynamics of </i>Cryptococcus neoformans</i> intracellular escape mechanisms," has been formally accepted for publication in PLOS Pathogens.

Best regards,

Kasturi Haldar

Editor-in-Chief

PLOS Pathogens

orcid.org/0000-0001-5065-158X

Michael Malim

Editor-in-Chief

PLOS Pathogens

orcid.org/0000-0002-7699-2064